# Global divergence in critical income for adult and childhood survival: analyses of mortality using Michaelis–Menten

Ryan J Hum[1]*, Prabhat Jha[2,3], Anita M McGahan[4], Yu-Ling Cheng[1]

[1]Department of Chemical Engineering and Applied Chemistry and Centre for Global Engineering, University of Toronto, Toronto, Canada; [2]Dalla Lana School of Public Health, University of Toronto, Toronto, Canada; [3]Li Ka Shing Knowledge Institute, St Michael's Hospital, Toronto, Canada; [4]Rotman School of Management, University of Toronto, Toronto, Canada

**Abstract** Life expectancy has risen sharply in the last 50 years. We applied the classic Michaelis–Menten enzyme kinetics to demonstrate a novel mathematical relationship of income to childhood (aged 0–5 years) and adult (aged 15–60 years) survival. We treat income as a substrate that is catalyzed to increase survival (from technologies that income buys) for 180 countries from 1970 and 2007. Michaelis–Menten kinetics permit estimates of maximal survival and, uniquely, the critical income needed to achieve half of the period-specific maximum. Maximum child and adult survival rose by about 1% per year. Critical incomes fell by half for children, but doubled for men. HIV infection and smoking account for some, but not all, of the rising critical incomes for adult survival. Altering the future cost curve for adult survival will require more widespread use of current interventions, most notably tobacco control, but also research to identify practicable low-cost drugs, diagnostics, and strategies.

## Introduction

In the 20th century, global life expectancy rose by about 90 days per year, with much of that increase driven by substantial declines in childhood mortality (*Oeppen, 2002*; *Vallin and Meslé, 2009*).

In most low-income countries, death in middle age is increasing in relative importance as other causes decrease and the effects of smoking increase (*Jha, 2009*). Most adult deaths are due to non-communicable diseases, such as vascular, respiratory, or neoplastic diseases, with a significant minority due to HIV/AIDS, malaria, and tuberculosis (*Beaglehole et al., 2011b*). This has spurred calls for global efforts to tackle chronic diseases of adults akin to the already established global efforts to reduce childhood, maternal, and infectious disease deaths through the United Nations (UN) 2015 Millennium Development Goals (MDGs) (*Beaglehole et al., 2011a*).

In a classic paper, *Preston (1975)* offered an elegant analysis to establish a cross-sectional relationship between mean national life expectancy and national income per capita. The 'Preston curve' shows a positive relationship between national income levels and life expectancy in poorer countries but with smaller marginal returns at higher incomes. Preston noted an upward rise in the curve for three decades over the 20th century, leading to higher life expectancies for any given national income (defined as GDP per capita) level over time (*Figure 1A* is a reproduced graph of the Preston curve for the years 1960 and 2000). This upward rise has been attributed to the adoption of low-cost health technologies for child survival (*Mathers et al., 2008*; *United Nations, 2011*), such as immunization and oral rehydration, and to improvements in nutrition (*Fogel, 1997*), sanitation (*Cutler and Miller, 2004*), primary education for women (*Caldwell, 1979*; *Cutler et al., 2006*), or a combination of these variables (*Rodgers, 2002*). Indeed, some countries such as Costa

*For correspondence: ryan.hum@mail.utoronto.ca

**eLife digest** In 1975 Samuel Preston published a classic paper that showed life expectancy was related to national income. When plotted as a graph, with national income on the horizontal axis and life expectancy on the vertical axis, the Preston curve shows that an increase in national income leads to an increase in life expectancy, with the increases in life expectancy becoming proportionally smaller as income increases. Moreover, Preston showed that innovations in healthcare (such as vaccinations, public health education, and sanitation systems) were increasing the maximum life expectancy that can be achieved for any given national income (defined as GDP per capita): this can be seen by comparing the Preston curves from 1960 and 2000 shown in Figure 1A. Indeed, global life expectancy increased by about 25 years over the course of the 20th century, which suggests that the level of daily income needed to achieve a certain life expectancy should be falling over time.

To explore this in greater detail, Hum et al. have constructed a mathematical model to investigate the relationship between health and income across different age groups and income levels. They found that most of the gains in life expectancy for low- and middle-income countries have been achieved by reducing child mortality, with gains in life expectancy for adults being restricted mostly to high-income countries. The model, which is based on the mathematical equations used to describe the kinetics of enzymatic reactions, makes it possible to estimate the improvements of health that can be made over time, and also the level of income that is needed to achieve these improvements.

In particular, Hum et al. have established a new parameter, the critical income, which is the level of income needed to achieve half of the maximal health found in high-income countries for the year in question. Based on available data from over 150 countries, they found that critical incomes fell by half for children between 1970 and 2007, but doubled for adult males during the same period. The rise in critical income for adults was due partly to the HIV epidemic and increases in smoking in low- and middle-income countries, reflecting the growing problems presented by noncommunicable diseases. Hum et al. conclude that increasing the survival among adults will require increased use of proven cost-effective interventions, most notably tobacco control, plus new research to identify low-cost drugs, diagnostics, and other public health strategies.

Rica, Cuba, and Sri Lanka all achieve life expectancy levels greater than expected by their income (*Caldwell, 1986*).

Life expectancy has continued to increase globally, and this has led to the belief that less daily income is required to achieve a certain life expectancy level (*Casabonne and Kenny, 2012*). But these trends mask heterogeneity across age groups and income levels. In this article, we argue that the majority of gains in life expectancy for low- and middle-income countries have been achieved through reductions in child mortality while improvements in adult mortality have been restricted mostly to high-income countries. This has reduced inequality between countries for child survival, however, increased inequality for adult survival. The central objective of this article is to construct a mathematical model derived from the field of biochemistry/enzymology to analyze upward and lateral movements in the Preston curve. More specifically, we establish a new parameter 'critical income' (analogous to the Michaelis constant [$K_m$]) to investigate the trends in income levels associated with reductions in mortality over the past 40 years; this model is further applied to age-specific populations to elucidate child and adult contributions to improving life expectancy. We also estimate the impact of HIV/AIDS and smoking on adult critical income values.

## Mathematical models of health and income

The relationship between health and income has been described using a variety of empirical models, generally of the form (*Wagstaff and Van Doorslaer, 2000*)

$$h_i = f(y_i); \text{ where } f' > 0 \text{ and } f'' < 0, \quad [1]$$

where $h_i$ is a health indicator (such as life expectancy or survival rates) and $y_i$ is income, for country or unit $i$.

Preston, originally, characterized the relationship between life expectancy ($e_0$) at birth and gross domestic product per capita ('GDP') as a logistic function in the form

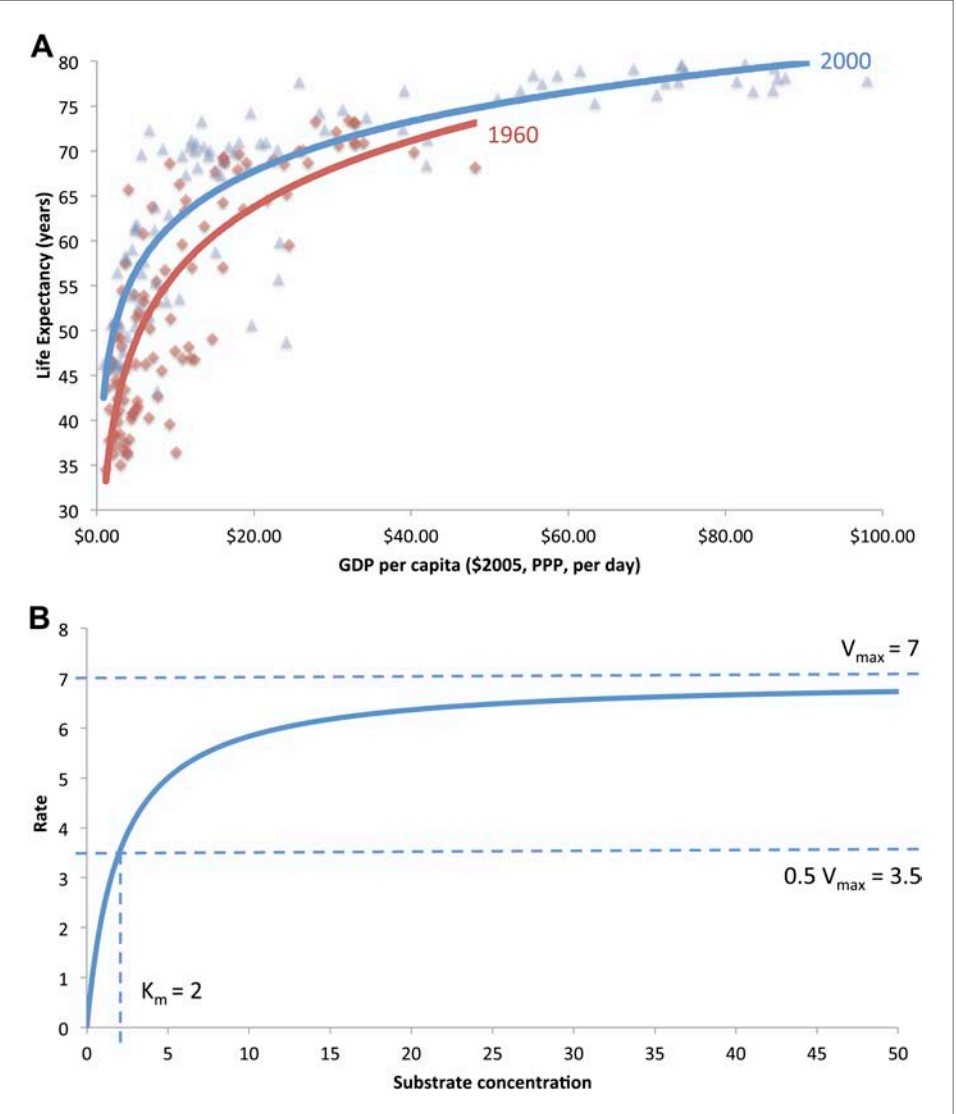

**Figure 1**. (**A**) The original 'Preston curve' (plotted as a logistic function) demonstrating an upward shift from 1930 to 1960. Source: *Preston (1975)*. (**B**) A hypothetical Michaelis–Menten plot with kinetic parameters $v_{max}$ and $K_m$.

$$e_0 = \frac{a}{1 + e^{\left(b + c \times d^{GDP'}\right)}}, \qquad [2]$$

where GDP' is GDP linearized on a scale of 0 to 1. *Figure 1A* is a reproduced plot of the original 'Preston curve', with an upward shift between the years 1960 and 2000 (*Preston, 1975*). Although the fit of this model to data is good, the model parameters are harder to interpret. It is suggested that identifying an inflection point (the point of diminishing returns) could clarify the analysis of these parameters (*Rogers and Crimmins, 2011*), for a generalized logistic function of the form

$$e_0 = \frac{c}{1 + a \times e^{(-b \times GDP)}}, \qquad [3]$$

where '$c$' is the maximal life expectancy and the inflection point is at a GDP value of '$\ln(a)/b$'.

More commonly, however, a log-linear relationship is used as a simpler alternative to the logistic function

$$e_0 = a + b \times \ln(GDP). \qquad [4]$$

The Preston model explains the majority of the observed variance in life expectancy, but the coefficients ('a' and 'b') are not interpretable as no meaning exists for each coefficient nor is a theoretical relationship between them stipulated. The inadequacy of the Preston models to methodologically identify a single point (such as an inflection point; *Rogers and Crimmins, 2011*), which describes the changing curvature at lower income levels, has led to greater emphasis on the upward rise in the Preston curve, and less attention is devoted to lateral movements (along the income axis). Nonparametric regressions have been used to identify a 'hinge' on the Preston curve (*Deaton and Case, 2009*); however, we know of no known study that has explicitly quantified shifts across income levels.

## Adapted enzyme kinetics model

Here, we identify a new construct called 'critical income' through the novel application of a mathematical model by Michaelis and Menten (MM) to empirically track child and adult mortality at different incomes. The Michaelis–Menten mathematical model first described in 1913 (*Michaelis and Menten, 1913*) was initially developed to analyze enzyme kinetics. Enzymes are biomacromolecules that act as catalysts, agents that accelerate the rate of a chemical reaction without being consumed in the process. In the absence of these enzymes, some thermodynamically favorable reactions may be kinetically hindered from occurring. For a single reaction, enzyme E binds to a substrate S to form an intermediate complex ES, which is converted into a product P and the original enzyme

$$E + S \rightleftharpoons ES \rightarrow E + P. \qquad [5]$$

*Figure 1B* is a plot of the MM equation for a hypothetical reaction. The classic MM equation describes the dependence of the enzyme velocity $v$ on substrate concentration $[S]$. $v$ asymptotically approaches a maximum value ($v_{max}$) at high $[S]$ when enzymatic sites are saturated. $K_m$ is the half saturation constant—substrate concentration at which $v = 0.5 v_{max}$. $K_m$ is a function of the forward and reverse reaction rate constants, where a lower $K_m$ value indicates a more efficient catalyst

$$v = \frac{v_{max}[S]}{[S] + K_m}. \qquad [6]$$

We extend the application of the Michaelis–Menten kinetics model to describe life expectancy, child, and adult survival. We use the analogy that GDP is a substrate, and health determinants and widespread applications of public health research, treatments and interventions are catalysts that increase health and survival (*Ad Hoc Committee on Health Research Relating to Future Intervention Options, 1996*). Indeed, infrastructure (such as water sanitation and education systems), vaccinations (leading to long-term immunity), and public health knowledge can be viewed as catalysts that are not consumed in entirety during the process.

$$e_0 = \frac{e_{0,max} \times GDP}{GDP + K_{inc}}, \qquad [7]$$

where for a given year, the mean life expectancy at birth ($e_0$) in a country is related to its GDP per capita (per day, at constant 2005 international dollars, adjusted for purchasing power parity and inflation). The MM model is characterized by two parameters: the life expectancy of the highest income countries ($e_{0,max}$), and critical income ($K_{inc}$). We introduce a new parameter called 'critical income' as a meaningful construct that can be estimated to assess the relationship between income and mortality. This construct is defined as the level of daily income associated with the achievement of 50% of the maximum life expectancy (i.e., GDP per capita at $e_0 = 0.5 \times e_{0,max}$); 'maximum life expectancy' is empiric, approximating the observed average life expectancy in high-income countries (*Rodgers, 2002*). While it is a biochemical convention to report the 50% mark, the critical income value is adaptable to determine higher fractional level of maximum life expectancy. In particular, two, four, and nine times the critical income yields the income required to achieve 66.7%, 80%, and 90% of the maximal life expectancy, respectively.

The asymptotic leveling of life expectancy at high income seen in the Preston or MM curves is analogous to the saturation of enzymatic sites at high substrate concentration, thus, further increases in GDP leads to only marginal increases in life expectancy.

This model establishes a systematic and empirical method for monitoring not only previously documented upward rises in the Preston curve but also potential shifts along the income axis. Whereas an upward rise can increase the maximum achievable survival for all countries, lateral shifts indicate an

increase or decrease of income required to achieve this survival. These lateral shifts are particularly significant for more resource-limited countries that have yet to reach asymptotic leveling in the health and income relationship.

Age-specific contributions to changes in life expectancy are quantified using child mortality rate as the probability of a child born in a specific year dying before reaching the age of 5 years, referred to as 5q0, and the gender-specific adult mortality rate, representing the probability (for a given year) that an individual who has just turned 15 years will not reach the age of 60 years, referred to as 45q15. We transform these values to survival rates, where a mortality rate of 5 per 1000 corresponds to a 99.5% survival rate. The corresponding MM functions for child survival from age 0 to 5 years (5p0) and for adults from the age of 15 to 60 years (45p15), by gender, are

$$5p0 = \frac{5p0_{max} \times GDP}{GDP + K_{inc}}; \quad [8]$$

$$45p15 = \frac{45p15_{max} \times GDP}{GDP + K_{inc}}, \quad [9]$$

where in a given year, $5p0_{max}$ and $45p15_{max}$ are the maximum survival rates in high income countries and $K_{inc}$ is the critical income associated with that age-specific group and gender.

Two large and widespread public health factors that have influenced mortality over the last four decades have been the HIV/AIDS pandemic and smoking, which makes more common most vascular, respiratory, and neoplastic diseases as well as tuberculosis (*Gajalakshmi et al., 2003*; *Jha, 2009*). We test if adjustment for the marked heterogeneity of HIV prevalence (between the ages of 15 and 49 years, as proxy for general population infection levels) and cigarette consumption (at ages 15 years or older) make less efficient the relationship between GDP per capita and adult mortality—as indicated by critical income values.

## Results

### Model fitness

*Figure 2A* shows the graphical similarity between the logistic, log-linear, and MM model fits of life expectancy for the year 1990. For the years 1970 to 2007, there was no significant difference in the coefficient of determination ($R^2$) for the logistic (M = 0.666, SD = 0.064), log-linear (M = 0.645, SD = 0.073), and MM regressions (M = 0.622, SD = 0.083; $F(2,26)$ = 0.807, p=0.458, not statistically significant), indicating that the MM model is statistically as valid as the Preston log-linear and logistic models. See *Table 1* for the regression coefficient for all years.

Although the fits of the models were comparable, the parameters of the logistic and log-linear models have less obvious explanatory value. For the logistic function, all inflection points were determined to be negative (from −$2.69 in 1970 to −$8.47 in 2007) and are impossible income values for any country. This suggests that the logistic function is overly complex and not necessary to model the data. For the log-linear function, the model parameters did not allow for a means to intuitively track lateral movements in the Preston curve. In addition, when using annual income (rather than daily income), the parameter '*a*' was not statistically significant for any year from 1970 to 2000. *Table 2* is a comparison of all three models for the year 1990.

For each year, the maximum life expectancy approximated the 5% trimmed mean life expectancy observed in countries with annual incomes greater than $12,276 ($t(8)$ = −1.596, p=0.149, ns; *Table 1*). In addition, we performed a sensitivity analysis on the maximum life expectancy and critical income parameters and determined that both were not sensitive to the random removal of 5% of the data ($t(8)$ = 1.07, p=0.314, not statistically significant; $t(8)$ = 1.46, p=0.178, not statistically significant, respectively; *Table 1*).

### Trends in life expectancy

An upward rise is observed for life expectancy and income from 1970 to 2007 (*Figure 2B*). The maximal life expectancy rose from 67.8 years (95% uncertainty interval 65.4–70.1 years) in 1970 to 75.5 years (95% uncertainty interval 73.9–77.1 years) in 2007. This change in maximal life expectancy represents a linear increase of 75 (95% uncertainty interval 48–99 years) days per calendar year ($R^2$ = 0.875) over

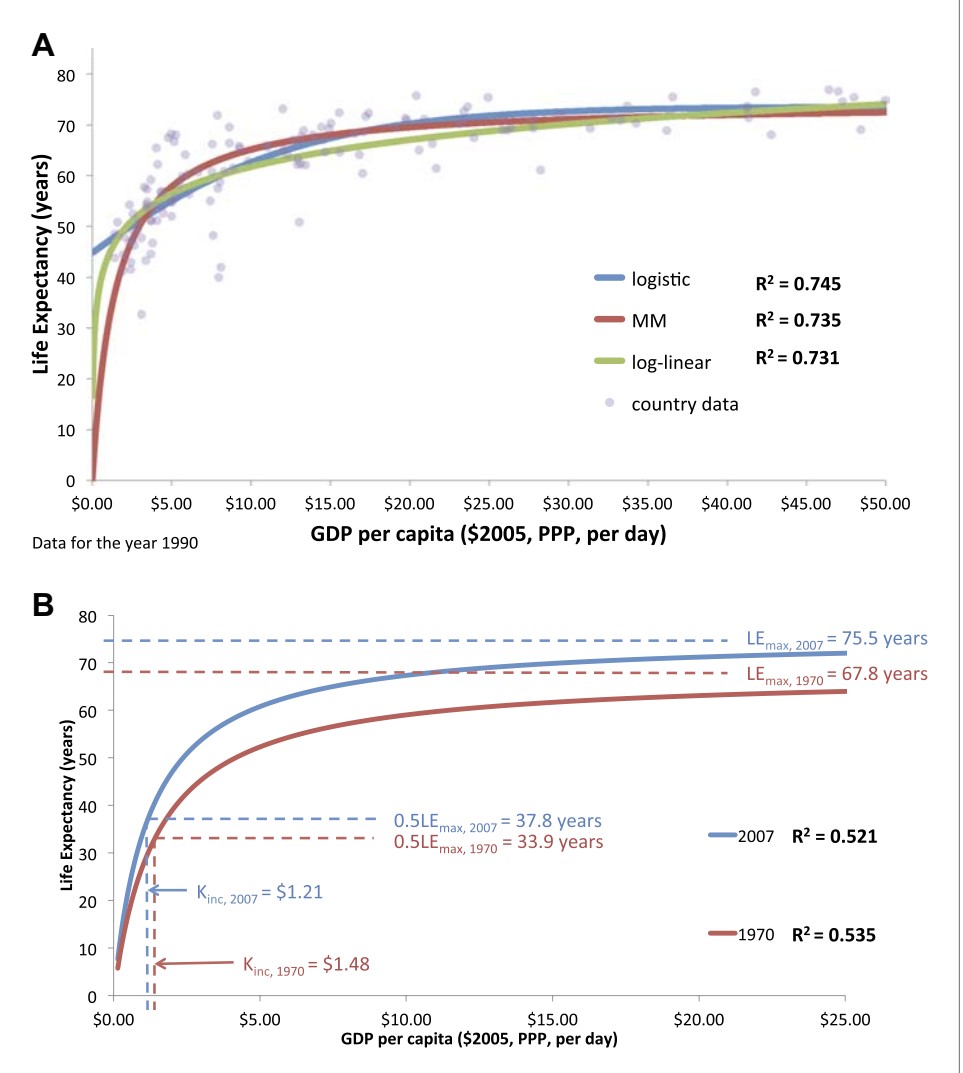

**Figure 2**. (**A**) The graphical similarity between the logistic, log-linear, and Michaelis–Menten model fits of life expectancy for the year 1990. (**B**) Preston curve plotted as an enzyme kinetics reaction with coefficients critical income and maximum life expectancy for the years 1970 and 2007.

the last 40 years, and is comparable to the life expectancy increase of almost 90 days per calendar year for the 20th century (**Oeppen, 2002**).

In addition to maximal life expectancy increase, a lower national income is associated with a higher life expectancy now than it was 40 years ago. The critical income (in constant 2005 international dollars) needed to achieve half of maximal overall life expectancy declined from $1.48 ($1.18–$1.78) in 1970 to $1.21 ($0.98–$1.44) in 2007, equivalent in 2007 to the extreme poverty line of $1.25 per day; this represents an 18% decrease in critical income to gain almost 4 additional years of life expectancy. Critical income declined linearly at a rate of −$0.09 per decade (−$0.06 to −$0.13, $R^2 = 0.839$). *Table 1* also reports the incomes required to achieve 66.7%, 80%, and 90% of the maximal life expectancies.

## Trends in adult and child survival

Maximum survival for all age groups rose at statistically the same rate between 1970 and 2007 (*Figure 3A*, *Table 3*). Maximum child survival to the age of 5 years increased from 94.5% (92.7–96.2%) to 98.0% (97.3–98.6%), an increase of 0.8% (0.3–1.3%) per decade ($R^2 = 0.665$); maximum adult female survival at ages 15–59 years rose from 85.9% (84.1–87.7%) to 90.1% (87.8–92.3%), an increase

**Table 1.** Maximum life expectancy, critical income, and regression coefficients (95% confidence intervals) for all countries at 5-year intervals from 1970 to 2007

| Year | n | $R^2$ | MaxLife expectancy (LEmax, years) | | 5% trimmed mean LE for high-income countries | Income require for varying levels of LEmax | | | | |
|---|---|---|---|---|---|---|---|---|---|---|
| | | | | | | Critical income ($K_{incr}$, 50%) | | | | |
| | | | Full sample | 95% random sample | | Full sample | 95% random sample | 66.70% | 80% | 90% |
| 1970 | 148 | 0.535 | 67.8 (65.4–70.1) | 67.6 | 66.7 | 1.48 (1.18–1.78) | 1.43 | 2.96 | 5.92 | 13.32 |
| 1975 | 148 | 0.574 | 69.3 (67.2–71.4) | 69.2 | 70.3 | 1.50 (1.22–1.77) | 1.53 | 3.00 | 6.00 | 13.50 |
| 1980 | 149 | 0.668 | 71.3 (69.6–73.0) | 71.1 | 71.8 | 1.51 (1.28–1.74) | 1.46 | 3.02 | 6.04 | 13.59 |
| 1985 | 152 | 0.716 | 73.2 (71.8–74.8) | 73.1 | 73.2 | 1.50 (1.29–1.70) | 1.46 | 3.00 | 6.00 | 13.50 |
| 1990 | 164 | 0.735 | 74.6 (73.2–75.9) | 73.5 | 74.3 | 1.45 (1.27–1.63) | 1.34 | 2.90 | 5.80 | 13.05 |
| 1995 | 177 | 0.677 | 75.0 (73.6–76.4) | 74.8 | 75.4 | 1.31 (1.13–1.49) | 1.27 | 2.62 | 5.24 | 11.79 |
| 2000 | 178 | 0.64 | 75.2 (73.8–76.7) | 75.3 | 76.2 | 1.27 (1.08–1.46) | 1.27 | 2.54 | 5.08 | 11.43 |
| 2005 | 177 | 0.532 | 75.3 (73.6–76.9) | 75.6 | 76.5 | 1.19 (0.97–1.41) | 1.23 | 2.38 | 4.76 | 10.71 |
| 2007 | 172 | 0.521 | 75.5 (73.9–77.1) | 75.7 | 76.4 | 1.21 (0.98–1.44) | 1.22 | 2.42 | 4.84 | 10.89 |

Note: All model parameters were found to be significant, p<0.0001.

of 1.1% (0.5–1.6%) per decade ($R^2$ = 0.762); and maximum adult male survival at ages 15–59 years rose from 77.7% (75.6–79.7%) to 82.1% (79.0–85.0%), an increase of 1.1% (0.6–1.7%) per decade ($R^2$ = 0.749).

However, critical incomes diverged dramatically for children and adults (**Figure 3B**). From 1970 to 1980, the critical income values for children, adult males, and adult females were statistically equivalent, between $0.54–$0.58 per day. For children, the critical income values declined gradually, however, with a large drop in 1990–1995. Over the 40-year period from 1970 to 2010, the critical income for child survival fell by over half from $0.58 ($0.46–$0.70) to $0.24 ($0.20–$0.28). In contrast, critical income more than doubled for adult male survival from $0.54 ($0.38–$0.71) to $1.20 ($0.81–$1.58) and rose over 50% for adult female survival from $0.57 ($0.43–$0.70) to $0.89 ($0.65–$1.12) (see **Table 3** for all years and **Supplementary file 1** for each country-specific critical income). These percentage increases were similar even if critical income was defined differently, for example, income needed to achieve 66.7%, 80%, or 90% of maximum survival (not shown).

The 1970 and 2007 survival curves for adult men crossed over at a value for $10.95 GDP per capita per day. There are 58 countries with a total adult male population of approximately 780 million (or 35% of the world adult male population) below this income value. For these countries, adult male survival was lower in 2007 than in 1970 (**Figure 4A**). For women, the comparable 1970 and 2007 survival curves crossed at a value of $5.93 GDP per capita per day, corresponding to 32 countries, with approximately 150 million women (or 7% of the world adult female population), where female survival in 2007 was worse than in 1970 (**Figure 4B**). By contrast, between 1970 and 2007, for the 58 countries with an income under $10.95 per capita per day in 2007, child mortality improved by 11.8%, adult male survival fell by 2.7%, and incomes rose by 8.9% (or an absolute increase of $0.98).

## Impact of HIV prevalence and smoking

Particularly given the marked increase in critical incomes needed to achieve maximal adult male survival, we tested if HIV infection and deaths and smoking (both, greater in males than in females) explained the increasing critical income values for adult males (**Figure 5**). For adult males in the year 2000, HIV prevalence (range 0.06–26% aged 15–49 years) and cigarette consumption (range 54.6–3385.2 per year at ages 15 years or older) were shown to influence critical income. Selecting the 86 countries with complete survival, income, HIV, and cigarette data, the critical income value was $2.02 ($1.32–$2.73). Adjusting for HIV and smoking prevalence to the average across countries reduced the critical income to $1.22 ($0.76–$1.69), meaning that HIV and smoking could explain about half of the increases in critical income. In 1970, the critical income was $0.54 ($0.38–$0.71), thus, HIV

**Table 2.** Comparison of the logistic adapted Michaelis–Menten and log-linear models for the year 1990

| Model | Form | R² | Parameters | | | |
|---|---|---|---|---|---|---|
| Logistic | $LE = \dfrac{LE_{max}}{a + e^{(-b \times GDP)}}$ | 0.745 | $LE_{max} = 73.6$ (72.1–75.1) | $a = 0.642$ (0.546–0.739) | $b = 0.129$ (0.159–0.100) | Inflection point = −3.43 |
| Adapted Michaelis–Menten | $LE = \dfrac{LE_{max} \times GDP}{(k_{inc} + GDP)}$ | 0.735 | $LE_{max} = 74.6$ (73.2–75.9) | $k_{inc} = 1.50$ (1.29–1.70) | | |
| Log-linear | $LE = a + b \times \ln(GDP)$ | 0.731 | $a = 44.1$ (42.0–46.2) | $b = 7.65$ (6.93–8.37) | | |

infection and smoking do not explain all of the worsening of critical income observed by the year 2000. HIV infection and smoking had no statistical impact on the maximum survival.

These results are further supported by a first differences analysis (*Table 4*). Over the 10-year period from 1990 to 2000, a rise of one cigarette consumed per day per person and 1% in HIV prevalence led, for adult men, to a $1.70 ($0.96–$2.43) and $0.70 ($0.27–$1.13) increase in critical income, respectively. For adult females, the impact of a 1% rise in HIV prevalence per capita was associated with a $0.40 ($0.31–$0.49) increase in critical income; however, the impact of smoking was lower, with a change of $0.38 ($0.23–$0.54). Moreover, adding a covariate for HIV prevalence also improved the goodness of fit dramatically for all years and genders (*Table 3*).

## Discussion

Our establishment of a new parameter 'critical income' provides novel insights into the relationship of global mortality changes with income. It builds on the well-established Preston curve functions by quantifying the rise in survival. More profoundly, our analyses reveal that while less and less income is required to improve childhood survival, the opposite is true to improve adult survival—particularly in low- and middle-income countries.

Unlike for adults, child survival replicates the upward and lateral trend in the original Preston curve for life expectancy. This suggests that for low- and middle-income countries, the majority of the past gains in life expectancy have been achieved via declining child mortality. This is consistent with the UN Population Division trends given the large impact of child mortality (compared to adult mortality) on overall life expectancy (*United Nations, 2011*). The greatest decline in child critical income was achieved after 1990, coinciding with actions following the UN's World Summit for Children. With justification, cost-effective interventions have been disproportionately devoted to child and maternal health (*Daar et al., 2007*), and more recently to control of infectious diseases such as HIV/AIDS, malaria, and tuberculosis. Increasing coverage of inexpensive health interventions such as immunization, insecticide-treated nets, and case management of childhood infections could be contributing to the decline in critical income for child survival (*Jamison et al., 2006*; *Mathers et al., 2008*). Moreover, there might be complementary benefits of education in reducing child mortality (*Gakidou et al., 2010*). Our study does not address any causal relationship between such interventions and reductions in child mortality; however, our results imply that the achievement of the UN MDG 4 (to reduce child mortality by two-thirds from 1990 levels) might be due to the falling levels of income needed to increase child survival.

For adult survival, however, there is a reversal of fortune. While achievable adult survival rates have improved, improvements are only associated with those countries at higher income levels. The rise in critical income suggests that the marginal costs of increasing adult longevity are rising; this may explain the lower rate of decline in adult mortality in countries with low income (*Rajaratnam et al., 2010*). The emergence of HIV/AIDS in the 1980s and the rise in global smoking prevalence in low- and middle-income countries (*Guindon and Boisclair, 2003*) can explain much, but not the entire rise in critical income for adult males. For males, the rise in smoking accounts for over 40% more of the variance in critical income compared to 30% for HIV/AIDS. For adult women, the impact of cigarettes on critical income is much lower, which reflects the five times lower prevalence of smoking among women compared to men (*Guindon and Boisclair, 2003*). Indeed, previous studies have already highlighted the impact of smoking and HIV/AIDS on adult survival in developing countries.

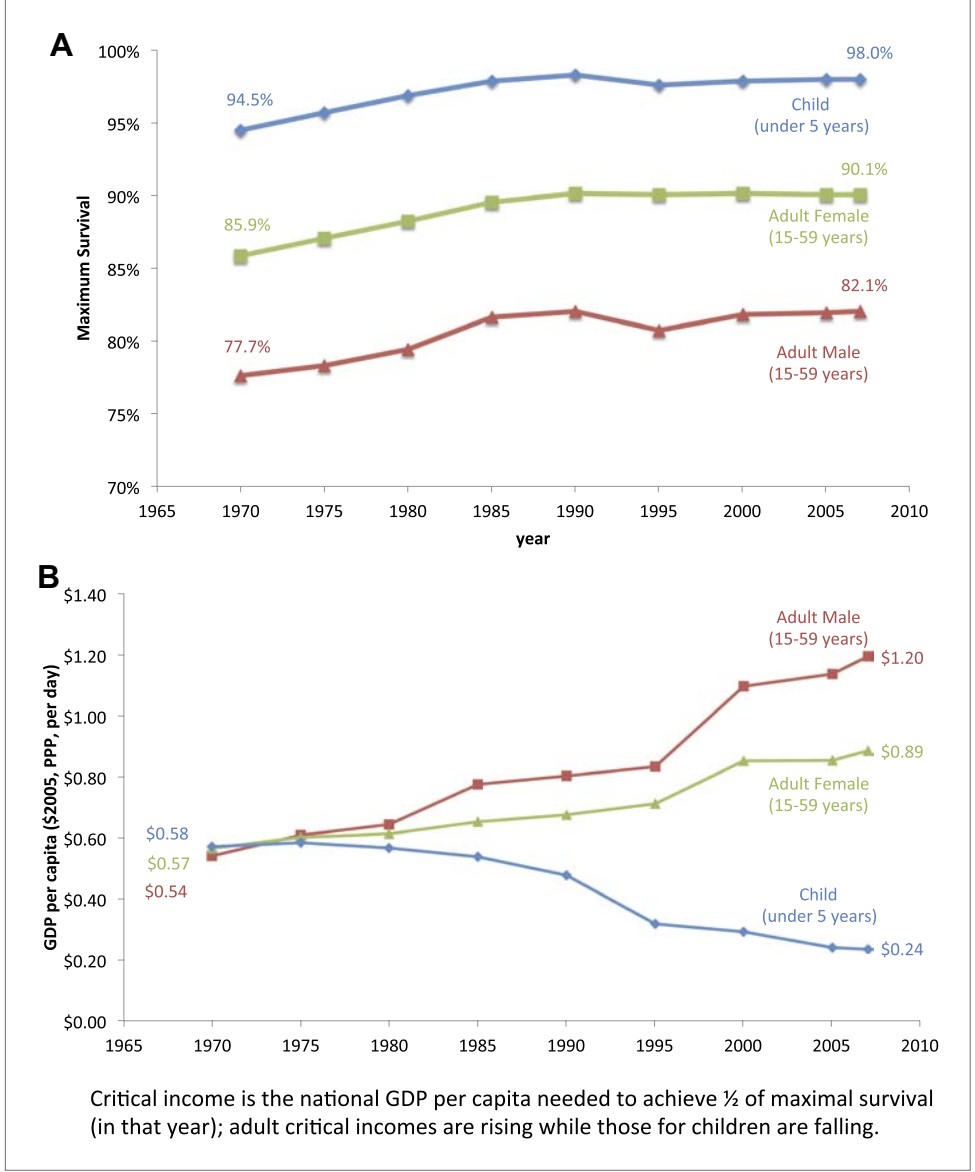

Critical income is the national GDP per capita needed to achieve ½ of maximal survival (in that year); adult critical incomes are rising while those for children are falling.

**Figure 3**. Trends for maximum survival (**A**) and critical income (**B**) for children and adults from 1970 to 2007.

Even low levels (4%) of HIV prevalence in rural Tanzania can increase overall adult mortality by more than 50% (***Todd et al., 1997***). Our findings also show that smoking increases critical income but has no statistical impact on maximum survival. This is in line with trends in global smoking, where prevalence of smoking (and subsequently the deaths attributed to smoking) are rising in low- and middle-income countries but declining in high-income countries (***Jha, 2009***).

Under the current conditions, an approximate national income per capita of $2.20 per day would be required in 2007 to attain the same achievable adult male survival rate with $1.25 per day in 1970. Moreover, should the critical income costs for adults continue to rise (in line with current trends), survival rates for low- and middle-income countries might well deteriorate into the future. In contrast, high-income countries have benefited from the rise in maximum survival among adults. This is likely due to more widespread availability of secondary treatments for chronic diseases, most notably for vascular disease, and in particular from sharp reductions in smoking (***Jha, 2009***).

The probabilities of premature adult deaths before the age of 70 from vascular, respiratory, and neoplastic diseases are remarkably similar in low-, middle-, countries and high-income countries

**Table 3.** Maximum survival, critical income, and regression coefficients (95% confidence intervals) from 1970 to 2005

| Year | Child | | | Female | | | Female (with HIV covariate) | | | | Male | | | Male (with HIV covariate) | | | |
|---|---|---|---|---|---|---|---|---|---|---|---|---|---|---|---|---|---|
| | R² | Max% | K$_{inc}$ $ | R² | Max% | K$_{inc}$ $ | R² | Max% | K$_{inc}$ $ | HIV | R² | Max% | K$_{inc}$ $ | R² | Max% | K$_{inc}$ $ | HIV |
| 1970 | 0.444 | 94.5 (92.7–96.2) | 0.58 (0.46–0.70) | 0.376 | 85.9 (84.1–87.7) | 0.57 (0.43–0.70) | | | | | 0.253 | 77.7 (75.6–79.7) | 0.54 (0.38–0.71) | | | | |
| 1980 | 0.569 | 96.9 (95.7–98.2) | 0.57 (0.48–0.66) | 0.448 | 88.3 (86.7–89.9) | 0.62 (0.49–0.74) | | | | | 0.286 | 79.5 (77.5–81.6) | 0.65 (0.46–0.83) | | | | |
| 1990 | 0.629 | 98.3 (97.4–99.2) | 0.48 (0.42–0.55) | 0.482 | 90.2 (88.7–91.8) | 0.68 (0.55–0.81) | 0.595 | 90.4 (88.8–91.4) | 0.54 (0.41–0.67) | −1.8 (−1.2 to −2.4) | 0.380 | 82.1 (80.2–84.1) | 0.81 (0.62–0.99) | 0.498 | 81.8 (79.8–83.8) | 0.60 (0.42–0.79) | −2.2 (−1.3 to −2.8) |
| 2000 | 0.530 | 97.9 (97.2–98.7) | 0.30 (0.25–0.34) | 0.377 | 90.2 (88.0–92.4) | 0.86 (0.65–1.06) | 0.787 | 92.1 (90.5–93.6) | 0.69 (0.55–0.82) | −1.8(−1.6 to −2.0) | 0.284 | 81.9 (79.0–84.8) | 1.10 (0.77–1.43) | 0.720 | 83.5 (81.4–85.6) | 0.79 (0.57–1.00) | −2.2 (−1.9 to −2.5) |
| 2005 | 0.466 | 98.0 (97.3–98.6) | 0.25 (0.20–0.29) | 0.323 | 90.1 (87.8–92.3) | 0.86 (0.63–1.12) | 0.803 | 92.4 (90.9–93.8) | 0.68 (0.54–0.81) | −2.2(−1.8 to −2.5) | 0.254 | 82.0 (79.0–85.0) | 1.14 (0.81–1.58) | 0.739 | 84.2 (82.1–86.3) | 0.82 (0.59–1.04) | −2.6 (−3.0 to −2.2) |

Note: All model parameters were found to be significant, p<0.0001.

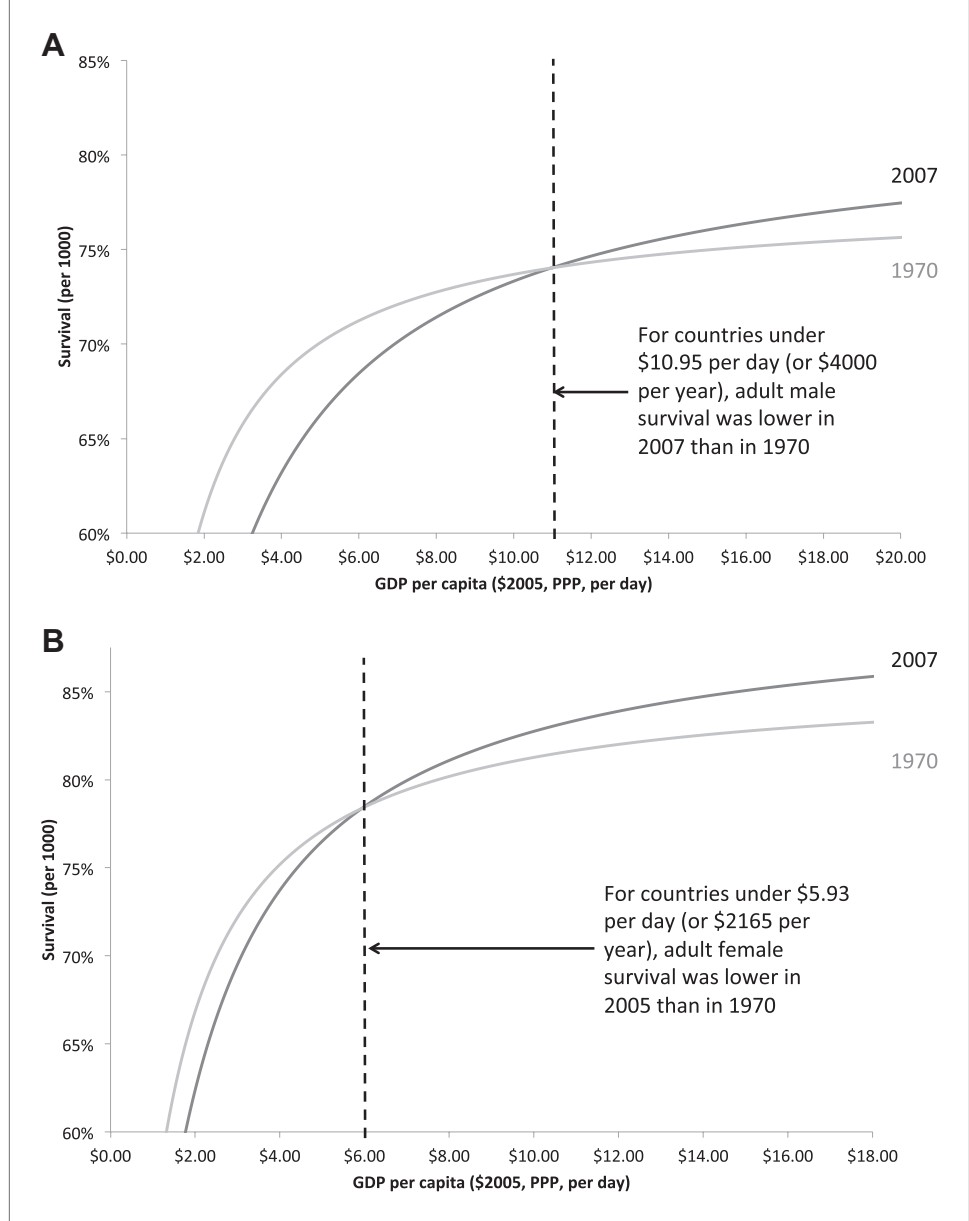

**Figure 4**. Adult male (**A**) and female (**B**) survival regression curves for the years 1970 and 2007. Adult survival in 2007 is lower than in 1970 for countries under the income threshold of $10.95 (for men) and $5.93 (for women). DOI: 10.7554/eLife.00051.009

(*Strong et al., 2005*). Low-cost, effective, and feasible interventions (*Jha et al., 2002*) exist against these diseases, most importantly tobacco control (*Jha, 2009*), but also increased reduction in hazardous alcohol intake (particularly in former Soviet states; *Zaridze et al., 2009*), and low-cost drugs for secondary management of existing disease. However, these interventions are still not widely used in low-income countries (*Jha et al., 2012*). Over the past few decades, research and development of new technologies (drugs, vaccines, and policies) have focused mostly on childhood and infectious disease, with fewer worldwide investments for adult chronic diseases (*Ad Hoc Committee on Health Research Relating to Future Intervention Options, 1996*). Thus, the longer-term trajectory of critical incomes for adult survival might well depend on the development of newer interventions, as well as more widespread application of interventions already proven to be cost effective (*Jamison et al., 2006*).

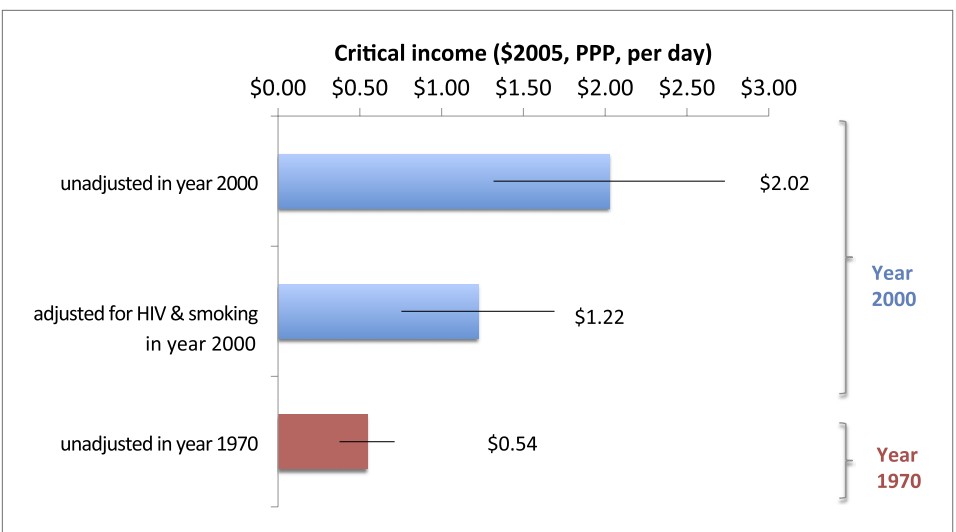

**Figure 5**. Impact of smoking and HIV on critical income for adult males in 2000.

This is, as far as we know, the first application of enzyme kinetics to mortality changes. There is no a priori reason to exclude a relationship of mortality and income being comparable to biological reactions. Indeed, since its initial formulation in 1913, the MM equation has been sufficiently adaptable to explain various levels of biological complexity from thousands of single enzyme reactions (***English et al., 2005***), reversible and quasi-steady state systems (***Briggs, 1925***), the growth of microbial cultures on a nutrient substrate (***Monod, 1949***), the growth in size of a variety of different animal species (***López et al., 2000***), and the ability of organisms to acclimate to changes in environmental conditions (***Bonachela et al., 2011***).

Like all models, ours faces certain limitations. The relationship of income to mortality is comparable using the log-linear, logistic, MM, or indeed other methods. But none of the models can directly elucidate the mechanism(s) that converts income into better survival. The novel insight from the MM method is that we can model income as a substrate that catalyzes further changes that more directly impact survival. National income is quite large in relation to health spending for most countries (the median national spending on health is 5% of the GDP, and the United States at about 18% of GDP represents an outlier). Thus, income fulfills, partially, the MM definition of a substrate that is an input converted into health interventions and their use. We also reason that income directly enables certain technologies, immunization programs, epidemiological knowledge, education, and sanitation systems and other areas, which may themselves be interpreted as 'catalysts'—agents that accelerate the rate of a reaction without being fully consumed in the process.

Moreover, the MM method advances some understanding of mechanisms by defining critical income levels that measure the efficiency of particular countries using available income to keep up with maximal achievable survival by other countries during that narrow 5-year time period. The new variable critical income (akin to $K_m$) provides a measure to observe a 'lateral shift' that was not possible mathematically with the log-linear or logistic models. Additionally, the obvious impacts of two large risk factors (HIV and tobacco) on adult survival suggest some mechanistic insights between income and survival (and indeed the fact that male survival was more unequal from greater male smoking, and female and male survival were affected equally by the more even spread of HIV infection, further strengthens this case).

We caution however that full explanation of the links between specific technologies catalyzed by income need more research. For example, the Monod equation (an adaptation of the MM equation to fit the growth of bacterial cultures; ***Monod, 1949***), which was first proposed in 1949, was only interpreted thermodynamically 50 years later (***Liu et al., 2003***). Nonetheless, this equation was (and continues to be) used extensively in the pharmaceutical and food industries as well as in waste treatment systems. Additionally, our analyses could be considerably strengthened by examining trends in age,

**Table 4.** First differences analysis for HIV prevalence and cigarette consumption on country-specific critical income from 1990 to 2000

| | N | R² | HIV α ($ per HIV %) | Standardized α | Smoking β ($ per cigarette per person per day) | Standardized β |
|---|---|---|---|---|---|---|
| Adult male | 92 | 0.240 | 0.70 (0.27–1.13) | 0.302 | 1.70 (0.96–2.43) | 0.425 |
| Adult female | 92 | 0.504 | 0.40 (0.31–0.49) | 0.655 | 0.38 (0.23–0.54) | 0.366 |

Note: All model parameters were found to be significant, p<0.0001.

gender, and cause-specific mortality, but cause-specific mortality data are simply unavailable for most countries; for instance, notwithstanding global efforts to identify data for child mortality, less than 3% of all child deaths worldwide were certified according to cause of death (*Liu et al., 2012*). Indeed, expanded efforts to measure causes of death is a big global priority (*Jha, 2012*; *Vogel, 2012*) and the development of new physically or biologically inspired analytical models can help elucidate better understanding of global mortality trends.

## Materials (data sources)

Data for 180 countries at 5-year intervals from 1970 to 2007 were included in the analysis, with definitions of income as per those from the World Bank (*Supplementary file 1*); the Institute for Health Metrics and Evaluation (*Institute for Health Metrics and Evaluation, 2012*), UN Population Division (*United Nations, 2011*), UNAIDS online database (*UNAIDS, 2010*), and Penn World Table 6.3 databases (*Heston et al., 2009*) were used for child and adult mortality, life expectancy and country population by age-groups, prevalence of HIV among adults aged 15–49 years, and GDP per capita at constant 2005 international dollars and purchasing price parity, respectively. Cigarette consumption per capita is from the American Cancer Society (*Guindon and Boisclair, 2003*). Availability of data for all 180 countries was not consistent across all years and all variables. Data from missing countries (by individual year and population group and disease and risk burden) were removed from that specific regression analysis. A list of countries is found in *Supplementary file 1*. The UN Population Division database for child (*UN Inter-agency for Child Mortality Estimates, 2012*) and adult (*United Nations, 2011*) mortality were used to confirm results. To confirm the completeness and accuracy of the Institute for Health Metrics and Evaluation database, statistical analysis was duplicated using 5-year rolling averages from UN Inter-agency Group for Child Mortality Estimates (*UN Inter-agency for Child Mortality Estimates, 2012*) and UN Population Division database adult mortality (*United Nations, 2011*), with general consistency in the trend lines (*Figure 6A,B*). 5-year rolling averages also increased the $R^2$ values compared to single-year regressions; for regressions using UN datasets, the average $R^2$ rose to above 0.5 for all age groups (including high HIV-prevalent countries)—compared with lower correlation coefficients for single-year regressions using the Institute for Health Metrics and Evaluation dataset (*Table 3*).

## Methods

For the Michaelis–Menten adapted model and the logistic model, a nonlinear regression analysis (using iterative parameter estimation algorithms; *Greco and Hakala, 1979*) was used to calculate the parameters. The logistic and log-linear regressions were analyzed for life expectancy to compare the goodness of fit. To study progress over time, we calculated these $e_{o,max}$, $5p0_{max}$ and $45p15_{max}$, and $K_{inc}$ values for each age group and gender at 5-year intervals from 1970 to 2007. Countries with missing mortality or income estimates were eliminated from the regression analysis on a year-by-year basis. We also computed estimates of critical income for each specific country, year, age group, and gender, assuming that the maximum survival was a constant across all countries. Additional analyses were completed by adding covariates for HIV prevalence in a given year (from 1990 to 2007) and cigarette consumption per capita (in the year 2000). For the year 2000, analysis was restricted to countries with complete data for both HIV prevalence and cigarette consumption (n = 86). First differences analysis

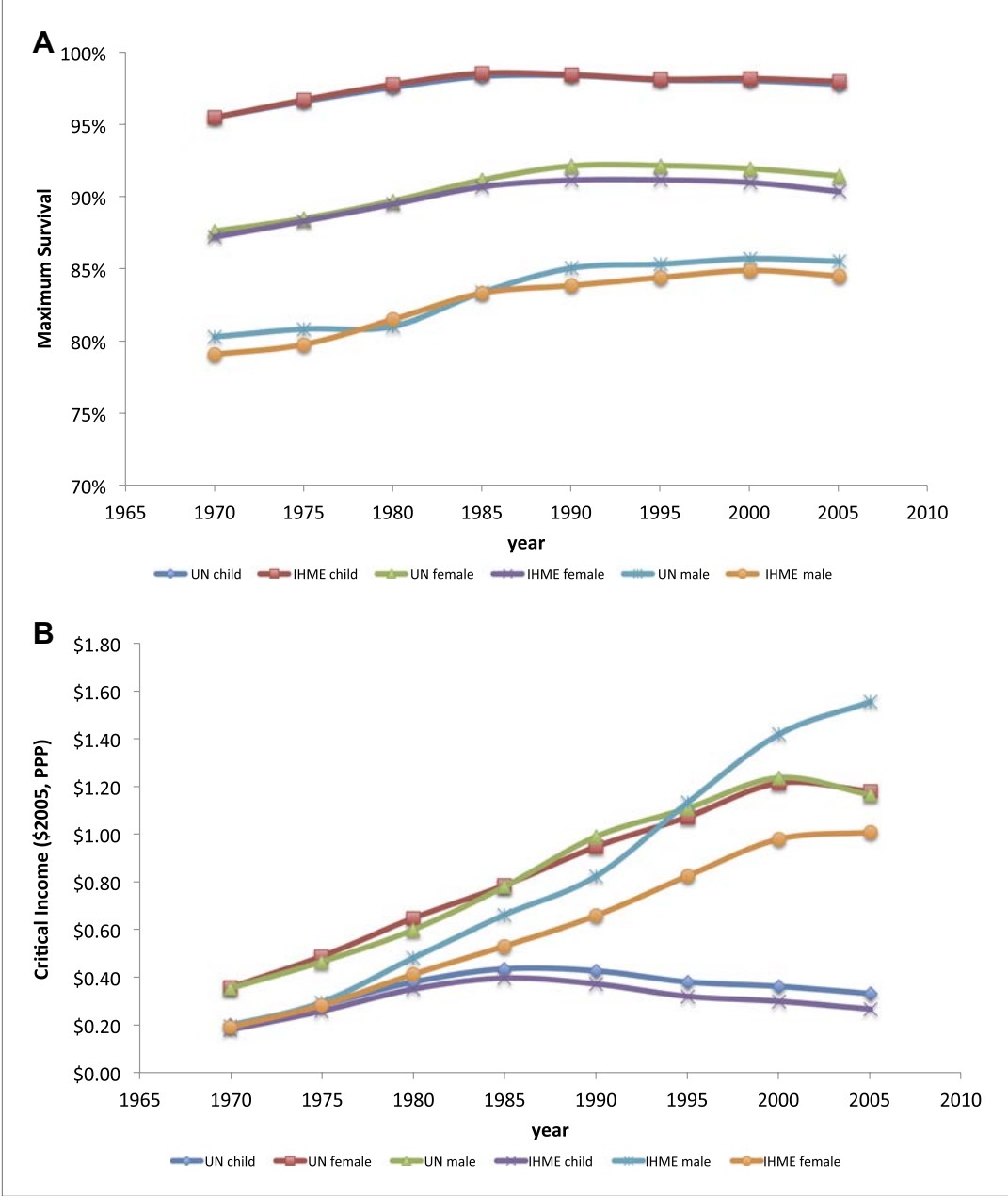

**Figure 6**. Child, adult male and adult female maximum survival (**A**) and critical income (**B**) curves from 1970 to 2005 using two different data sources (IHME and UN Population Division). Maximum survival and critical income values were calculated using 5-year averages where the national income per capita and country survival rates for year *i* was an average for the years *i* to *i* + 4.

was used to confirm the impact of HIV/AIDS prevalence and cigarette consumption on critical income values over the 10-year period from 1990 to 2000, such that

$$K_{inc} = f(HIV, Cig), \qquad [10]$$

$$\Delta K_{inc} = \alpha \Delta HIV + \beta \Delta Cig; \quad \text{where} \quad \alpha = \frac{\partial K_{inc}}{\partial HIV} \quad \text{and} \quad \beta = \frac{\partial K_{inc}}{\partial Cig}. \qquad [11]$$

We used SPSS (version 19) to conduct all regression analyses.

## Acknowledgements

We thank Beverly Bradley for comments.

## Additional information

### Competing interests

PJ: Senior Editor, *eLife*. The remaining authors have no competing interests to declare

### Funding

| Funder | Grant reference number | Author |
|---|---|---|
| Canadian Institute for Health Research | | Ryan J Hum |
| Lupina Foundation | | Ryan J Hum |
| National Sciences and Engineering Research Council | | Yu-Ling Cheng |
| Social Sciences and Humanities Research Council | | Anita M McGahan |
| Disease Control Priorities 3 | | Prabhat Jha |
| Metcalfe Fellowship | | Ryan J Hum |

The funders had no role in study design, data collection and interpretation, or the decision to submit the work for publication.

### Author contributions

RJH, Conception and design, Acquisition of data, Analysis and interpretation of data, Drafting or revising the article; PJ, Conception and design, Analysis and interpretation of data, Drafting or revising the article; AMM, Analysis and interpretation of data, Drafting or revising the article; Y-LC, Conception and design, Analysis and interpretation of data, Drafting or revising the article

## Additional files

### Supplementary files

- Supplementary file 1. Country specific critical incomes for adult male, adult female and children (in constant $2005 with purchasing price parity) for the years 1970, 1990 and 2007.

### Major datasets

The following previously published datasets were generated:

| Author(s) | Year | Dataset title | Dataset ID and/or URL | Database, license, and accessibility information |
|---|---|---|---|---|
| Institute for Health Metrics and Evaluation | 2010 | Adult Mortality Estimates by Country 1970–2010 | http://www.healthmetricsandevaluation.org/ghdx/record/adult-mortality-estimates-country-1970-2010 | Publicly available |
| UN Inter-agency for Child Mortality Estimates | 2012 | Under-five mortality rate estimates 1931–2011 | http://www.childmortality.org/files_v9/download/CME%20all%20countries%20U5MR%20estimates.xlsx | Publicly available |
| UNAIDS | 2010 | Estimated HIV Prevalence (Ages 15–49) | http://www.unaids.org/en/dataanalysis/datatools/aidsinfo/ | Publicly available |
| Heston A, Summers R, Aten B | 2009 | Penn World Table 6.3 database | https://pwt.sas.upenn.edu/php_site/pwt63/pwt63_form.php | Publicly available |
| Guindon GE, Boisclair D | 2003 | Past, Current and Future Trends in Tobacco Use, Appendix 6, Per Capita Cigarette Consumption 1970–2000 | http://escholarship.org/uc/item/4q57d5vp | Publicly available |

| United Nations, Department of Economic and Social Affairs, Population Division | 2011 | 45q15—Male | http://esa.un.org/unpd/wpp/Excel-Data/mortality.htm | |
| United Nations, Department of Economic and Social Affairs, Population Division | 2011 | 45q15—Female | http://esa.un.org/unpd/wpp/Excel-Data/mortality.htm | |
| Institute for Health Metrics and Evaluation | 2010 | Infant and Child Mortality Estimates by Country 1970–2010 | http://www.healthmetricsandevaluation.org/ghdx/record/infant-and-child-mortality-estimates-country-1970-2010 | Publicly available |
| United Nations, Department of Economic and Social Affairs, Population Division | 2011 | Life Expectancy at Birth—Both Sexes | http://esa.un.org/unpd/wpp/Excel-Data/mortality.htm | Publicly available |

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
