## [Decision Letter]

Thank you for choosing to send your work entitled “Global divergence in critical income for adult and childhood survival: analyses of mortality using Michaelis-Menten” for consideration at *eLife*. Your article has been evaluated by a Senior Editor, a Reviewing Editor, and by 3 external peer reviewers. The Reviewing Editor has assembled the following comments based on the reviewers' reports.

The reviewers' two key concerns are about your use of the Michaelis-Menten analogue and whether your conclusion about the potential role of HIV infection and smoking on the final impact of income on health is really justified. These concerns are articulated below. We expect you will be able to address them in your revision.

**Comments relating to the Michaelis-Menten function:**

* The use of the MM function to model the relationship between national income and health needs to be better justified. In what way is national wealth converted into better health by the action of an “enzyme”, i.e. something (measured by v-max) limited in quantity but not itself used up in the process?

* If the MM function is not used for mechanistic reasons, and is just a convenient form of function to explain the relationship between GDP and life expectancy, then it needs to be justified statistically. This is problematic because it appears to fit just as well (or badly) as the log-linear model. There is no attempt to show that the log-linear model is a worse fit to countries at low-income levels.

* K_inc is intuitively easy to explain but is a problematic indicator because virtually all the data points are above it. Hence it may not be a very robust parameter to estimate, and is also difficult to visualise what a country with such a low-income level may look like. It might be better to take the daily level of income associated with achievement of the median life expectancy (for example). It would be good also to explore how sensitive to data the conclusions are about this critical income level, e.g., by removing certain data points.

* The authors should explain more clearly why the more classic logistic function (or one of the many S-curve variations are either inappropriate or do not have the adequate properties.

* Since the model generated by Michaelis-Menten dynamics clearly reaches an asymptote, one needs to fit it to a function that converges to an asymptote. There are essentially two ways to do this: either 1) use a converging exponential a*(1-exp(-b*x)) where a would be the asymptote, be the rate of increase and x would be GDP, or something similar with an extra parameter such as a logistic curve, or 2) use a polynomial function a*x/(1+b*x) where a, b, and x would play more or less the same role, or again something similar with an extra-parameter which could involve powers. What would make the study more interesting would be if one could either a) choose between the several possible functions on the basis of the data or b) choose between the several possible functions on the basis of a theoretical understanding of what is going on.

**Comments relating to health significance:**

* The important influence of HIV and smoking sounds very plausible, but is not incorporated into the model in a systematic way. A better way would be to take the list of top 5 or 10 (say) childhood mortality causes for which there are adequate data, and incorporate them all into the model, then remove the ones that are less important by some variable selection technique. Currently, it isn't clear whether HIV and/or smoking are just proxy variables for some other causes of death, or even the general amount of effort placed on public health issues in a given country.

* There have been dramatic changes in the three leading risk factors for the global disease burden over the last 20 years. Currently these are high blood pressure, tobacco smoke, and alcohol use compared to 1990, when the top 3 risk factors were childhood underweight, household air pollution from solid fuel use, and tobacco smoke. This has been very well described in a recent analysis in *the Lancet*. Income alone does not adequately explain these changes – rather it is a reflection of a range of complex interactions of which GDP and family income are two factors. It is unclear to us how this analysis deals with the complex web of interactions that impact on life expectancy or the burden of disease and our read of the manuscript did not leave us with a clear indication of how this enzyme mathematical model adds value as a new approach to the problem if it does not deal with the multitude of variables impacting life expectancy.

---

## [Author Response]

We believe our paper has benefited from the reviewer comments. To address the specific comments:

**1. Comments relating to the Michaelis-Menten function:**

*a. The use of the MM function to model the relationship between national income and health needs to be better justified. In what way is national wealth converted into better health by the action of an “enzyme”, i.e. something (measured by v-max) limited in quantity but not itself used up in the process*?

Response: We have expanded the discussion of limitations of applying the MM to model the relationship between national income and health. First, we cannot attribute a theoretical mechanism to our model; but we reason that income directly enables certain technologies, immunization schemes, public health knowledge, education, and sanitation systems, which may be interpreted as “catalysts”, whose effects are long lasting, are not consumed in the reaction (or on minimally consumed as they cost only a very small % of GDP), and can increase the efficiency of the transformation of income to health. Second, the main contribution of our finding is to identify a new variable (akin to *K*_*m*_), which provides a measure of efficiency or “lateral shift” that simply was not possible mathematically from the earlier adaptations of the Preston curve. This critical income value is also directly policy relevant – it shows marginal costs are falling for children but rising for adult survival.

Historically, the literature has often identified a relationship but determined the mechanism much later. For example, Jacques Monod who in 1949 adapted the MM equation from single enzymes to fit the growth of bacterial cultures: his work was only interpreted thermodynamically 50 years afterwards.

These points are now added to the methods and discussion.

*b. If the MM function is not used for mechanistic reasons, and is just a convenient form of function to explain the relationship between GDP and life expectancy, then it needs to be justified statistically. This is problematic because it appears to fit just as well (or badly) as the log-linear model. There is no attempt to show that the log-linear model is a worse fit to countries at low-income levels*.

Response: Notwithstanding the merits of the analogy, we agree that the Preston curve can be modeled empirically using a variety of mathematical functions with a converging asymptote. To that end, we have expanded our analysis in the manuscript to include regressions using a logistic, log-linear and MM model. (See subsequent comment replies.) We have removed the earlier confusing citations regarding a worse fit for low-income countries.

We have also clarified that the MM is not simply a different ways of showing the relationship between GDP and survival (and that the usual tests of statistical fitness are as robust as alternatives), but that it provides a new metric to capture lateral shifts or efficiency – that is critical income.

*c. K_inc is intuitively easy to explain but is a problematic indicator because virtually all the data points are above it. Hence it may not be a very robust parameter to estimate, and is also difficult to visualize what a country with such a low-income level may look like. It might be better to take the daily level of income associated with achievement of the median life expectancy (for example). It would be good also to explore how sensitive to data the conclusions are about this critical income level, e.g., by removing certain data points*.

Response: We agree that the 50% value is a relatively low value and that not many countries have such a low income level. However, we believe strongly in retaining this as: (a) it’s conceptually simpler; (b) it makes the mathematics and their future use, say applied to specific diseases, easier. That being said, the critical income value is adaptable to determine higher fractional level of maximal life expectancy. For example, two-, four- and nine-times the critical income would yield the incomes associated with 66.7%, 80%, and 90% of the maximal life expectancy, respectively. The proof that four times the critical income yields the income (GDP) required to achieve 80% of the maximal life expectancy is below:e0=45vmax=vmax∗GDPGDP+kinc4(GDP+kinc)=5 ∗ GDPGDP=4 ∗ kinc

In addition, we show that our MM model asymptote (LE_max_) approximates the 5% trimmed life expectancy of high-income countries. We also perform a sensitivity analysis on the model parameters by randomly removing 5% of the countries in the dataset; we show that the critical income and maximal life expectancy are not significantly impacted.

*d. The authors should explain more clearly why the more classic logistic function (or one of the many S-curve variations) are either inappropriate or do not have the adequate properties*.

Response: See response above. For the logistic model, we find the fitness to be statistically equivalent to that of the MM analogue; however, the interpretability of the model parameters is less intuitive given that all the inflection points were irrational (negative) income values. For the log-linear regressions, without transforming the income from annual to “per day” values, many of the model parameters were not statistically significant.

Given the fitness of the model and the additional tests, we believe this model generates a unique, easy to generate and extremely useful parameter in critical income that is not found in the other log-linear and logistic models.

*e. Since the model generated by Michaelis-Menten dynamics clearly reaches an asymptote one needs to fit it to a function that converges to an asymptote. There are essentially two ways to do this: either 1) use a converging exponential a*(1-exp(-b*x)) where a would be the asymptote, be the rate of increase and x would be GDP, or something similar with an extra parameter such as a logistic curve, or 2) use a polynomial function a*x/(1+b*x) where a, b, and x would play more or less the same role, or again something similar with an extra-parameter which could involve powers. What would make the study more interesting would be if one could either a) choose between the several possible functions on the basis of the data or b) choose between the several possible function on the basis of a theoretical understanding of what is going on*.

Response: Thank you. We have included further analysis in our manuscript regarding the logistic, log-linear and MM functions. We did not however add the exponential function (a*(1-exp(-b*x), as this function resulted in a statistically worse fit (R^2^ mean =0.452, SD=.106) for the data compared to the MM function (t(8)=-14.5, p<0.0001). In addition, the polynomial function a*x/(1+b*x) is algebraically identical to the MM function, where a=LEmax/Kinc and b=1/Kinc.

**2. Comments relating to health significance:**

*a. The important influence of HIV and smoking sounds very plausible, but is not incorporated into the model in a systematic way. A better way would be to take the list of top 5 or 10 (say) childhood mortality causes for which there are adequate data, and incorporate them all into the model, then remove the ones that are less important by some variable selection technique. Currently, it isn't clear whether HIV and/or smoking are just proxy variables for some other causes of death, or even the general amount of effort placed on public health issues in a given country*.

Response: Our objective was two-fold. First, to provide the MM construct and show it is empirically robust. The second was to test which are the determinants of Critical Income. The choice of HIV/AIDS and tobacco is obvious as these are large and still growing causes of death for adults (Todd et al. HIV-associated adult mortality in a rural Tanzanian population. AIDS. 1997May11;11(6):801.

Jha P. Avoidable global cancer deaths and total deaths from smoking. Nat Rev Cancer. 2009 Aug.20;9(9):655–64). Moreover, there are gender differences, which are shown below.Smoking prevalence is rising in low and middle income countries and declining in high-income countriesSmoking increases critical income and has no statistical impact on maximum survivalSmoking prevalence is higher in men than womenSmoking has a greater impact on critical income for adult males compared to adult femalesHIV prevalence is highest in low and middle income countriesHIV prevalence increases critical income and has no statistical impact on maximum survivalHIV prevalence impacts men and women equallyHIV prevalence decreases critical income values in both men and women.

We see future extension of the MM testing alternative interventions, and these will be subject to better cross sectional data for countries as well as better information not only on overall survival but death rates from specific conditions (say cancers, heart disease, etc).

*b. There have been dramatic changes in the three leading risk factors for the global disease burden over the last 20 years. Currently these are high blood pressure, tobacco smoke and alcohol use compared to 1990, when the top 3 risk factors were childhood underweight, household air pollution from solid fuel use and tobacco smoke. This has been very well described in a recent analysis in the Lancet. Income alone does not adequately explain these changes – rather it is a reflection of a range of complex interactions of which GDP and family income are two factors. It is unclear to us how this analysis deals with the complex web of interactions that impact on life expectancy or the burden of disease and our read of the manuscript did not leave us with a clear indication of how this enzyme mathematical model adds value as a new approach to the problem if it does not deal with the multitude of variables impacting life expectancy*.

Response: While we believe our model could be amended to include other risk factors, many of the existing databases with time series (such as fasting plasma glucose level, blood pressure, body-mass index, and cholesterol), are relatively new and all explicitly use income as a covariate to create their estimates. Moreover, the data available are only of all cause deaths. Cause-specific time series data for children or adults are not yet available to enable the kind of analyses proposed by the reviewer. We welcome further efforts to “count the dead”, given for example only 3% of all child deaths worldwide are certified according to cause of death (Liu et al. 2012).

**References**

Danaei et al., 2011. National, regional, and global trends in fasting plasma glucose and diabetes prevalence since 1980: systematic analysis of health examination surveys and epidemiological studies with 370 country-years and 2·7 million participants. *The Lancet* - 2 July 2011 (Vol. 378, Issue 9785, Pages 31–40)

Danaei et al., 2011. National, regional, and global trends in systolic blood pressure since 1980: systematic analysis of health examination surveys and epidemiological studies with 786 country-years and 5·4 million participants. *The Lancet* - 12 February 2011 (Vol. 377, Issue 9765, Pages 568–577)

Finucane et al. 2011. National, regional, and global trends in body-mass index since 1980: systematic analysis of health examination surveys and epidemiological studies with 960 country-years and 9·1 million participants. *The Lancet* - 12 February 2011 (Vol. 377, Issue 9765, Pages 557–567)

Farzadfar et al. 2011. National, regional, and global trends in serum total cholesterol since 1980: systematic analysis of health examination surveys and epidemiological studies with 321 country-years and 3·0 million participants. *The Lancet* - 12 February 2011 (Vol. 377, Issue 9765, Pages 578–586)

Liu et al. 2012. Global, regional, and national causes of child mortality: an updated systematic analysis for 2010 with time trends since 2000. *The Lancet* - 9 June 2012 (Vol. 379, Issue 9832, Pages 2151–2161)